# CryoSPIN: Improving Ab-Initio Cryo-EM Reconstruction with Semi-Amortized Pose Inference

**Shayan Shekarforoush**[1, 2]
shayan@cs.toronto.edu

**David B. Lindell**[1, 2]
lindell@cs.toronto.edu

**Marcus A. Brubaker**[1, 2, 3, 4]
mab@eecs.yorku.ca

**David J. Fleet**[1, 2, 4]
fleet@cs.toronto.edu

[1]University of Toronto    [2]Vector Institute    [3]York University    [4]Google DeepMind

## Abstract

Cryo-EM is an increasingly popular method for determining the atomic resolution 3D structure of macromolecular complexes (eg, proteins) from noisy 2D images captured by an electron microscope. The computational task is to reconstruct the 3D density of the particle, along with 3D pose of the particle in each 2D image, for which the posterior pose distribution is highly multi-modal. Recent developments in cryo-EM have focused on deep learning for which amortized inference has been used to predict pose. Here, we address key problems with this approach, and propose a new semi-amortized method, cryoSPIN, in which reconstruction begins with amortized inference and then switches to a form of auto-decoding to refine poses locally using stochastic gradient descent. Through evaluation on synthetic datasets, we demonstrate that cryoSPIN is able to handle multi-modal pose distributions during the amortized inference stage, while the later, more flexible stage of direct pose optimization yields faster and more accurate convergence of poses compared to baselines. On experimental data, we show that cryoSPIN outperforms the state-of-the-art cryoAI in speed and reconstruction quality. Project Webpage.

## 1 Introduction

Single particle electron cryo-microscopy (cryo-EM) has gained popularity among structural biologists as a powerful experimental method for determining the 3D structure of macromolecular complexes, such as proteins and viruses, unlocking our understanding of biological function at the scale of the cell. Thanks to pioneering advances in hardware and data processing techniques, cryo-EM has enabled reconstruction of challenging structures at atomic or near-atomic resolution [1].

During a cryo-EM experiment, $10^4$–$10^7$ particle images, each containing an instance of the target bio-molecule, are acquired using a transmission electron microscope. From that *particle stack*, the goal is to reconstruct the unknown 3D density map [2]. This *ab-initio* reconstruction task presents some challenges. First, the 3D pose (orientation and position) of the particle in each image is unknown, and must be inferred along with the 3D structure. Second, low electron exposures are used to limit radiation damage to the particles (e.g., see Fig. 1). But this reduces the signal-to-noise ratio (SNR), obscuring high-resolution image detail and hindering pose and structure estimation. Third, bio-molecules are typically non-rigid and exhibit structural variations within a sample. Hence, for such heterogeneous datasets, it is crucial to account for the conformational variability in order to obtain high-resolution reconstruction [3–6].

38th Conference on Neural Information Processing Systems (NeurIPS 2024).

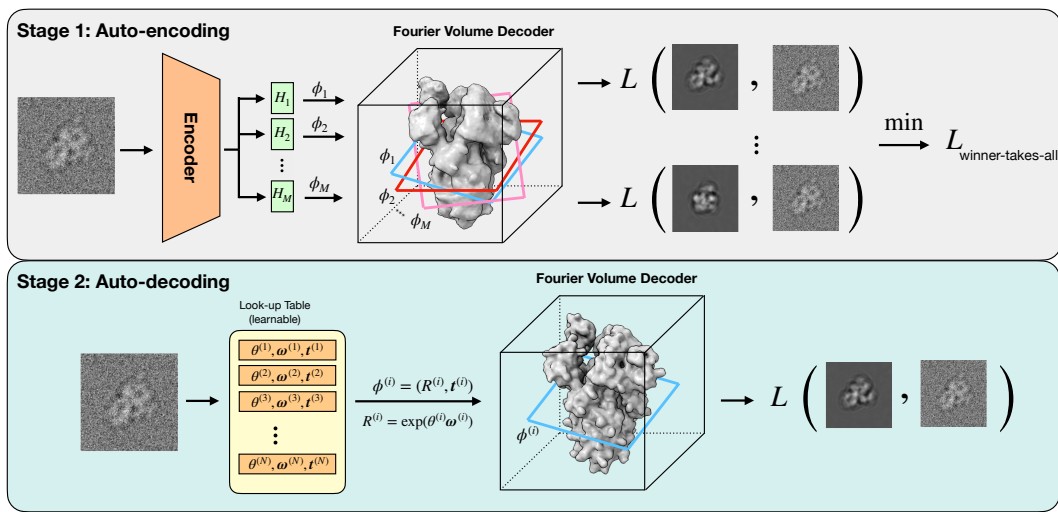

Figure 1: CryoSPIN consists of two stages: (i) an auto-encoding stage where an image encoder equipped with multiple heads maps the input image to the pose candidate set $\{\phi_1, \ldots \phi_M\}$, followed by computing projections by slicing through the volume decoder in Fourier space based on the pose set. These projections are compared with the input image and the one with the minimum error is used. (ii) An auto-decoding stage where pose parameters in axis-angle representation are stored for all images. The same volume decoder is used to obtain projections, and the reconstruction loss is computed for a single projection.

The development of methods in cryo-EM has recently focused on deep learning. CryoDRGN [5] proposed an image encoder-volume decoder architecture to model continuous heterogeneity, but the poses were assumed to be known. CryoDRGNv2 [7] introduced hierarchical pose search, comprising grid search followed by a form of branch-and-bound (BnB) optimization, akin to cryoSPARC [8]. More recently, cryoPoseNet [9] and cryoAI [10] introduced amortized pose inference, to avoid the expense of orientation matching for each particle image. They employ a convolutional neural network (CNN) to map an input image to a pose estimate. Nevertheless, although efficient, amortized inference may not accurately represent multi-modal posterior pose distributions, which occur frequently early in reconstruction before the structure is estimated well. As such, compared to previous methods like cryoSPARC [8] and cryoDRGNv2 [7] that consider many pose candidates, amortized methods sometimes fail to identify the correct mode. Moreover, amortization confines the pose search to a global parametric function (the encoder) which can lead to slow convergence.

Here, we introduce a new approach, called cryoSPIN, to ab-initio homogeneous reconstruction that is able to handle multi-modal pose distributions with a tailored encoder, and accelerates pose optimization with semi-amortization [11]. First, we perform amortized pose inference through a shared multi-choice encoder that maps each input image to multiple pose estimates (Fig. 1, top). In contrast to cryoPoseNet [9] and cryoAI [10], which use pose encoders that produce one or two estimates, we attach multiple pose predictor heads to a shared CNN feature extractor to predict multiple plausible poses for each image. By design, our multi-output encoder is able to account for pose uncertainty and encourage exploration of pose space during the initial stages of reconstruction. For the 3D decoder, unlike cryoDRGN and cryoAI that use MLPs with computationally expensive feed-forward implicit networks, we adopt an explicit parameterization to further accelerate the reconstruction. To train the encoder-decoder architecture, we apply a "winner-takes-all" loss in which the 3D decoder is queried to obtain a 3D-to-2D projection for each predicted pose. Inspired by the loss function in Multi-Choice Learning [12–14], we select the projection with the lowest image reconstruction error to determine the loss.

During the course of optimization, as higher resolution details are resolved, the pose posterior becomes predominantly uni-modal. Thus, the pose search can be narrowed down to a neighborhood containing the most likely mode. In this stage, we propose to transition from auto-encoding to auto-decoding (Fig. 1, bottom). In particular, for each image, we choose the pose with the lowest reconstruction loss, iteratively refine it with stochastic gradient descent (SGD), and continue pose optimization

and reconstruction in an alternating fashion. This direct, per-image optimization procedure achieves faster convergence to more accurate poses than amortized inference alone, which relies on potentially sub-optimal predictions of the encoder as a parametric function.

We evaluate cryoSPIN in comparison to state-of-the-art methods, cryoSPARC [8], cryoAI [10], and cryoDRGN [7] on both synthetic and real experimental datasets. Using synthetic data, we find that semi-amortized inference noticeably accelerates the convergence of poses, and yields 3D reconstructions at similar or higher resolutions than baseline methods. We also show that our multi-choice encoder empirically accounts for multiple modes in the pose posterior. Finally, we apply cryoSPIN to an experimental dataset and obtain reconstructions with higher resolution than those from cryoAI, while matching the resolution of cryoSPARC reconstructions.

To summarize, we make following contributions.

- Building upon cryoAI, we develop a new encoder based on a multi-head architecture to return multiple plausible candidates to further mitigate pose ambiguity.
- We train the encoder coupled with an explicit volume decoder using a *winner-take-all loss*, which penalizes the pose hypothesis with least reconstruction error.
- Beyond the architecture and objective, we introduce *semi-amortized pose inference*, which begins with feed-forward amortized pose inference, followed by direct, per-particle pose optimization.
- In ab-initio reconstruction, our semi-amortized method, cryoSPIN, is faster than amortized method of cryoAI and achieves higher-resolution on synthetic and experimental datasets.

## 2   Related work

**Cryo-EM reconstruction.**   Methods for cryo-EM reconstruction can be categorized as either homogeneous or heterogeneous. Homogeneous techniques [8–10] assume a rigid structure while heterogeneous ones [5, 7, 6, 15] allow for conformational variation. We focus on homogeneous reconstruction, but our optimization framework could be extended to heterogeneous data as well.

Early reconstruction techniques rely on common-lines [16–18] or projection mapping [19, 20] to select optimal poses. Other works  [21, 22] frame the reconstruction problem in the context of maximum a posteriori (MAP) estimation, and jointly reconstruct pose and structure via expectation maximization (EM). We compare our approach to cryoSPARC [8], a state-of-the-art method that uses stochastic gradient descent and a branch-and-bound search [23] for ab initio reconstruction and pose estimation. Like these methods, our auto-decoding stage directly optimizes pose of every image.

More recently, amortized inference techniques have been proposed for pose estimation [24, 9, 10, 15]. These techniques avoid explicit per-particle pose optimization; instead, they train an auto-encoder or variational one [25] to associate each particle image with a predicted pose [9]. One challenge is that the auto-encoders can become stuck in local optima during training [9]. To address this issue, cryoAI [10] produces two pose estimates per image coupled with a symmetrized loss function that penalizes the best one. We build on this concept by adopting a multi-head neural architecture as the encoder to output multiple plausible pose candidates and avoid local optima.

**Multi-choice learning (MCL).**   Inspired by scenarios where a set of hypotheses needs to be generated to account for uncertainty in the prediction task, MCL [12] was introduced in a supervised setup to learn multiple structured-outputs with SSVMs [26]. Their motivating question was: *can we learn to produce a set of plausible hypotheses?* To address this, they define an"oracle" loss in which only *the most accurate* output pays the penalty. This loss is minimized even if there is only a single accurate prediction in the set. The early follow-up work [27, 13] uses the same loss to learn a deep CNN ensemble composed of $M$ heads with a shared backbone network. Importantly, they show that the ensemble-mean loss hurts prediction diversity across different heads, while training with the "oracle" loss yields specialized heads. Variations have since been proposed to mitigate hypothesis collapse or overconfidence issues in MCL by modifying the loss or applying learnable probabilistic scoring schemes [28–30, 14]. MCL has been used to mitigate the ambiguity in several tasks including image segmentation [31], optical-flow estimation [32], trajectory forecasting [33], human pose and shape estimation [14, 34]. In our work, we use the "oracle" loss in context of auto-encoder which supervises the pose encoder indirectly through projections provided by the decoder.

# 3 Problem Definition

Image formation in cryo-EM is well-approximated using the weak-phase object model [35]. Under this model, the structure of interest is an unknown 3D density map, $V : \mathbb{R}^3 \to \mathbb{R}_{\geq 0}$, represented in some canonical coordinate frame. Cryo-EM images, $\{I_i\}_{i=1}^N$, are approximated as orthographic projections of the 3D map under some unknown coordinate transformation. We denote the unknown orientation by a rotation $R_i \in SO(3)$ and the unknown position in terms of an in-plane translation $t_i = (t_x, t_y) \in \mathbb{R}^2$. Formally,

$$I_i(x,y) = [g_i \star (S_{t_i} \mathcal{P}_{R_i} V)](x,y) + n(x,y) , \tag{1}$$

where $\mathcal{P}_{R_i}(\cdot)$ is the linear operator computing the integral along the optical axis, $z$, over the input density map rotated by $R_i$, and $S$ is the shift operator. The projection is convolved with the image-specific point-spread function (PSF), $g_i$, and corrupted by additive noise $n$. It is common to assume that $n$ follows a zero-mean white (or colored) Gaussian distribution.

By the Fourier slice theorem [36], the Fourier transform of a projection is equal to a central slice through the density map's 3D Fourier spectrum. That is,

$$\hat{I}_i(\omega_x, \omega_y) = \hat{g}_i \hat{S}_{t_i} (\hat{\mathcal{P}}_{R_i} \hat{V})(\omega_x, \omega_y) + \hat{n}(\omega_x, \omega_y) , \tag{2}$$

where $\hat{I}$ and $\hat{V}$ denote the 2D and 3D Fourier transforms of the image and the density map. The slice perpendicular to the projection is computed by $(\hat{\mathcal{P}}_{R_i} \hat{V})$. The translation by $S_{t_i}$ becomes a phase shift operator $\hat{S}_{t_i}$, and convolution with $g_i$ is equivalent to element-wise multiplication with, $\hat{g}_i$, the contrast transfer function (CTF). The noise $\hat{n}$ remains zero-mean Gaussian.

Under this model, given the structure $\hat{V}$, the negative log-likelihood of observing image $\hat{I}_i$ with noise variance $\sigma^2$ and pose $(R_i, t_i)$ is

$$\mathcal{L} = -\frac{1}{2\sigma^2} \sum_{\omega_x, \omega_y} [\hat{g}_i \hat{S}_{t_i} (\hat{\mathcal{P}}_{R_i} \hat{V})(\omega_x, \omega_y) - \hat{I}_i(\omega_x, \omega_y)]^2 . \tag{3}$$

Ab-initio reconstruction methods [8–10, 7] solve jointly for the unknown structure $\hat{V}$ and per-particle poses $(R_i, t_i)$. They often follow an Expectation-Maximization (EM) [37, 21] procedure in which the E-step aligns images with the structure yielding pose estimates $(R_i, t_i)$, and then in the M-step the volume $\hat{V}$ is updated by minimizing the negative log-likelihood in Eq. 3. Since errors in pose estimates lead to blurry reconstructions, accurate pose estimates are crucial to finding high-resolution structures. As discussed above, poses are either optimized through search and projection matching [8, 7] or estimated by an encoder network [10, 9, 15].

# 4 Methodology

We introduce cryoSPIN, a two-staged approach to ab-initio cryo-EM reconstruction, combining auto-encoding and auto-decoding. We start with amortized inference (Fig. 1, auto-encoding) where a shared encoder equipped with multiple heads provide a set of pose guesses. Multiple guesses enable multimodal inference in the initial stages of reconstruction when the 3D structure is poorly resolved. As reconstruction improves, the pose posterior becomes less uncertain, at which point we switch to direct pose optimization (Fig. 1, auto-decoding). The former enables handling the pose uncertainty early in reconstruction while the latter circumvents sub-optimality of encoder network leading to arguably more accurate poses and faster convergence. We couple our pose estimation module with an explicit volumetric decoder representing the 3D structure in Hartley space [5]. Our explicit model enables faster evaluation of projections compared to multiple passes through implicit neural representations [38–40]. The following sections discuss these components in detail.

## 4.1 Multi-choice encoder

With randomly initialized or a low-resolution reconstruction, the pose posterior contains multiple modes. Also, due to high levels of noise in images and near-symmetries in biological structures, pose estimation inherently involves high uncertainty. As a result, there exist several plausible poses for each image, and naive optimization or search is prone to local minima.

To account for uncertainty in pose estimation, we build upon cryoAI [10] and extend the encoder to return multiple candidate poses $\phi$. Formally, given the image $I_i \in \mathbb{R}^{H \times W}$, $M$ pairs of rotations and translations are computed as,

$$\phi_{i,j} = (R_{i,j}, t_{i,j}) = H_{\theta_j}(F_i), \quad 1 \leq j \leq M \tag{4}$$

where $R_{i,j} \in SO(3)$ and $t_{i,j} \in \mathbb{R}^2$. The image-specific intermediate features, $F_i = \text{VGG}(I_i) \in \mathbb{R}^{C \times H \times W}$ are extracted by a shared convolutional backbone based on the VGG16 architecture [41], which are then supplied to $M$ separate pose predictor heads, $H_{\theta_j}, 1 \leq j \leq M$, yielding poses $\phi_{i,j}$. The heads are composed of fully-connected layers with distinct learnable weights, while the backbone is shared across all heads. Using multiple heads facilitates pose exploration and reduces uncertainty in the pose estimation process. Moreover, as $\theta_j$ are randomly initialized, different heads are free to specialize on subsets of pose space so that they collectively produce a set of likely poses $\{\phi_{i,1}, \ldots, \phi_{i,M}\}$.

Inspired by multi-choice learning [12–14], we optimize the encoder-decoder using a "winner-take-all" loss. For each image $I_i$ and predicted pose $\phi_{i,j}$, the negative log-likelihood, $\mathcal{L}_{i,j}$ in (Eq. 3), is computed, and the minimum is selected as the final loss for the corresponding image, i.e.,

$$\mathcal{L}_i = \min_j \mathcal{L}_{i,j} . \tag{5}$$

Minimizing this loss requires only one of the predicted poses to be accurate. Interestingly, we find that this approach produces heads that specialize to somewhat disjoint regions of pose space (see Supp. C), consistent with previous work showing that it improves diversity in prediction tasks [13, 27]. This 'divide-and-conquer' approach facilitates pose space exploration, which is crucial in order to avoid local minima during the early stages of reconstruction where uncertainty in the 3D structure is significant. CryoAI [10] can be viewed a special case of this formulation; it assigns two poses to each image as a consequence of input augmentation, and selects the best one with a symmetrized loss. In contrast, our approach augments the output of the encoder network with multiple heads, each providing a pose estimate.

## 4.2 Switching from auto-encoding to auto-decoding

As optimization progresses and fine-grained details of the 3D structure are resolved, the posterior pose variance tends to decrease, with the posterior approaching a unimodal distribution. At this point, the gap between the amortized and variational posterior distributions is mainly determined by the error in the pose estimate (predicted mean), prioritizing accuracy over exploration. However, a feed-forward network, as a globally parameterized function of the input image, may be limited in the accuracy of its predictions, and hence the approximations inherent in amortized inference become significant sources of error in the refinement of the 3D structure. In prior work [11, 42–44], a related issue, called the *amortization gap*, is characterized by the KL-divergence between the true and predicted (amortized) variational posteriors.

To address this issue, we adopt a semi-amortized inference scheme [11] comprising two stages. First, the encoder predicts a set of pose candidates using a multi-head architecture. In the second stage, rather than amortized inference, pose parameters are directly optimized for each image using stochastic gradient descent. To initialize poses for the $i$-th image, we choose the one with the lowest reconstruction loss from the set of candidates $\{\phi_{i,1}, \ldots, \phi_{i,M}\}$, namely $\phi_i^* = \phi_{i,s}$ such that

$$s = \arg\min_j \mathcal{L}_{i,j} . \tag{6}$$

Subsequently, the pose and structure are optimized by coordinate descent using the negative log-likelihood (Eq. 3) as the objective function. See Supp. A for more details on rotation optimization.

## 4.3 Explicit representation as volume decoder

Recent works [10, 5] use coordinate networks [39, 40, 38] to implicitly model the Fourier representation of the 3D structure. Instead, we couple the pose estimation module with an explicit parameterization (3D voxel array) of the structure in the Fourier domain. The explicit representation is less computationally expensive than an MLP to evaluate and update. Also, this choice is motivated by the fact the implicit decoder needs to be queried multiple times for each image with the multi-head encoder.

We parameterize the volume using the Hartley representation [5]. The Fourier and Hartley transforms, respectively denoted as $F(\omega)$ and $H(\omega)$, are related as

$$H(\omega) = \mathcal{R}[F(\omega)] - \mathcal{I}[F(\omega)], \tag{7}$$

where $\omega$ denotes the frequency coordinate and $\mathcal{R}$ and $\mathcal{I}$ are the real and imaginary part, respectively. The Hartley representation is real-valued, and so more memory efficient to use than storing complex-valued Fourier coefficients. To account for high dynamic range of the Hartley coefficients, we assume the Hartley field is decomposed into mantissa, $m(\omega)$ and exponent $e(\omega)$ fields [10] as,

$$H(\omega) = m(\omega) \times \exp(e(\omega)). \tag{8}$$

This decomposition restricts the range of values for $m(\omega)$ and $e(\omega)$ and makes the reconstruction less sensitive to the initialization of the field.

## 5  Experiments

In what follows, we consider both synthetic and real datasets to empirically compare our semi-amortized ab-initio reconstruction method with state-of-the-art methods, cryoAI [10] and cryo-DRGN [7] which use amortized inference for reconstruction, and cryoSPARC [8] which performs pose search separately for each particle image.

**Synthetic data.**   We generate synthetic datasets by simulating the image formation process formalized in Sec. 3 using atomic models deposited in the Protein Data Bank (PDB). We compute ground-truth density maps of size $128^3$ for a heat shock protein (HSP) [45] (1.5 Å), pre-catalytic spliceosome [46] (4.33 Å), and SARS-CoV-2 spike protein [47] (2.13 Å). Then, $N = 50,000$ projections of size $L = 128$ are generated by randomly rotating and projecting the density map along the canonical z-axis. To simplify the analysis and interpretation of results (e.g. Figs. 4, 5), we consider synthetic particles are centered, and therefore omit the estimation of 2D translation and allowing us to visualize the pose posterior more easily. Finally, a random CTF is applied in the Fourier space and Gaussian noise is added yielding SNR $= 0.1$.

**Experimental data.**   As a widely adopted experimental benchmark, we use the 80S ribosome dataset (EMPIAR-10028 [48]), containing 105,247 particle images of linear box size $L = 360$ with pixel size 1.34 Å. Following cryoAI and cryoDRGN, we downsample the images to $L = 128$ (3.76 Å) using cryoSPARC software [8], and randomly split the data into two halves and run the reconstruction methods independently on each (to enable Fourier Shell Correlation as a quantitative measure of the resolution of the 3D reconstruction). Unlike synthetic data, particles are not well-centered so it is required to estimate translation parameters as well as rotations. This is achieved by augmenting each head to return translation parameters as well.

**Implementation details.**   During auto-encoding, we use encoders with $M = 7$ and $M = 15$ heads for reconstruction on synthetic and real datasets, respectively, with Adam [49] to optimize encoder and decoder with learning rates 0.0001 and 0.05. Once switched to direct optimization (after 7 epochs for synthetic and 15 epochs for real data), we reduce the decoder learning rate to 0.02 and allocate a new optimizer for pose parameters with learning rate 0.05. To be consistent with cryoAI, we use a batch size of 64 and train for the same number of epochs (20 for synthetic, 30 for real data). We use the public cryoAI codebase, run cryoSPARC v4.4.0 [8] with default settings, and run cryoDRGN homogenous ab-initio job. Methods are implemented in Pytorch [50]. Experiments are run on a single NVIDIA A40 GPU.

### 5.1  Results

We first qualitatively compare final reconstructions of cryoSPIN with cryoAI [10], cryoDRGN [7], and cryoSPARC [8] on synthetic and real datasets (Fig. 2, left). We observe that cryoAI gets stuck in local minima when evaluated on experimental data (eg, 80S ribosome). In particular, we found that in the presence of unknown planar shift, cryoAI fails to estimate effective translation parameters, but after centering all particles via pre-processing, we could reproduce the cryoAI results. Hence, the provided results for cryoAI in Fig. 2 are obtained after this pre-processing step, whereas our method and others are given off-centered experimental images. Both cryoSPIN and cryoSPARC capture high-frequency details of the 3D structure on all datasets, whereas reconstructions by amortized

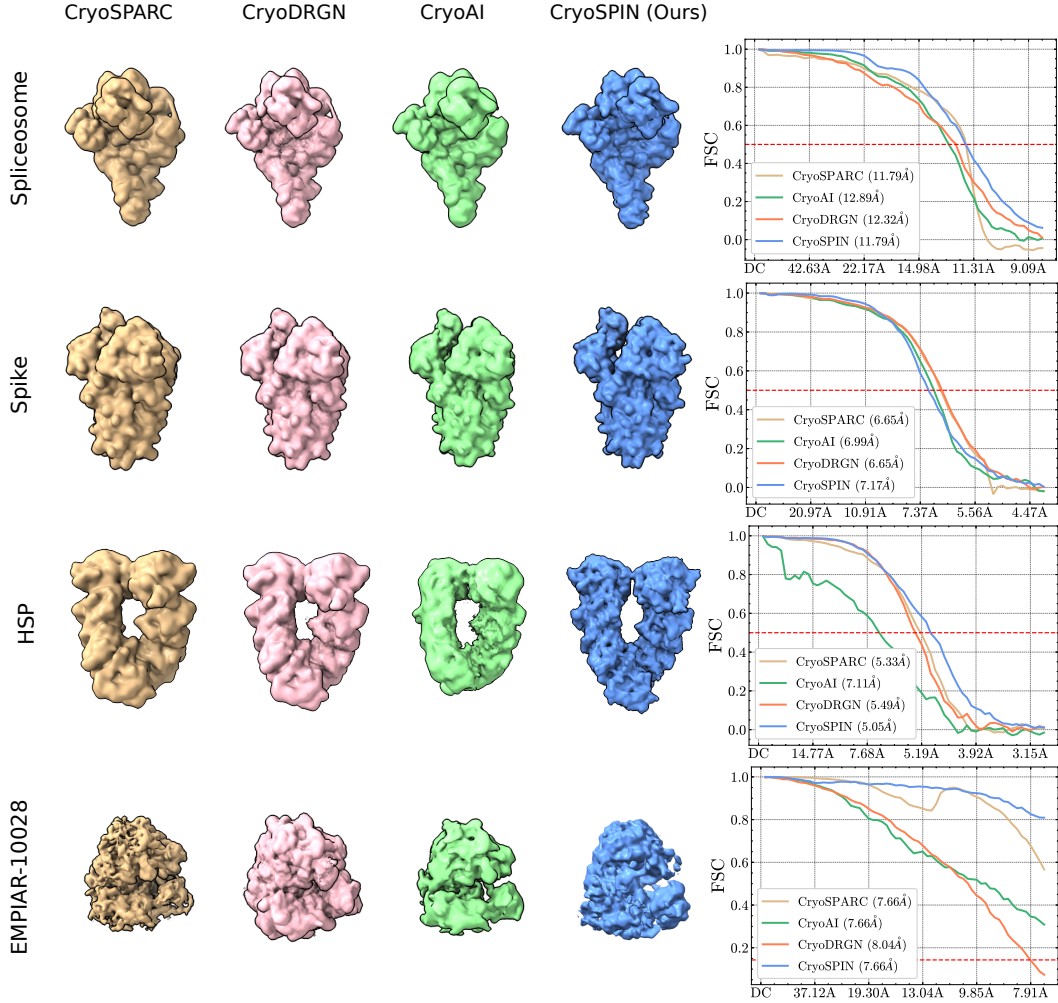

Figure 2: Qualitative and quantitative comparison of reconstructions obtained by our proposed semi-amortized method, cryoSPIN, with cryoAI [10], cryoDRGN [7] and cryoSPARC [8] We note that cryoAI often becomes stuck in local minima when particles are not centered, so for cryoAI we translate the input images to correct for the spatial offset. **(Left)** Final 3D reconstructions on three synthetic datasets and one experimental dataset (EMPIAR-10028) are depicted using ChimeraX [51]. **(Right)** FSC curves are visualized for quantitative comparison. The red dashed lines show the standard threshold levels of $0.5$ and $0.143$ to report the resolution (in Angstrom) for synthetic and real data, respectively. CryoSPIN achieves higher resolution on the Spliceosome and HSP datasets, and it is competitive with the state of the art on the Spike and EMPIAR-10028 datasets.

methods, cryoAI and cryoDRGN, are inferior on the HSP and 80S datasets. In particular, on HSP, cryoAI gets stuck in local minima as it fails to handle high uncertainty in poses caused by symmetries in this structure.

For quantitative comparison, we visualize the Fourier Shell Correlation (FSC) [52] between the reconstruction and the ground truth (Fig. 2, right). FSC is the gold-standard metric measuring the normalised cross-correlation coefficient between two 3D Fourier volumes along shells of increasing radius. Higher FSC implies more accurate reconstruction. On the Spliceosome, HSP and 80S experimental datasets, our reconstruction outperforms amortized methods of cryoAI and cryoDRGN. Also, our method outperforms cryoSPARC on Spliceosome, HSP and 80S, while achieving competitive FSC on Spike. We use the standard $0.5$ and $0.143$ cutoff thresholds to report the resolution for synthetic and real data, respectively. CryoSPIN achieves higher or competitive resolution compared to others on all datasets.

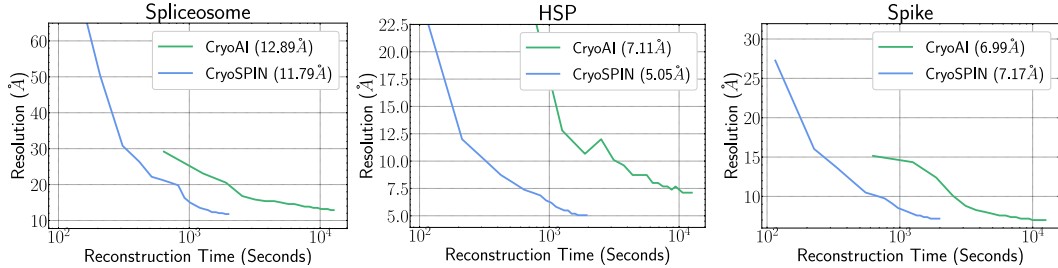

Figure 3: 3D resolution as a function of log time for different methods. These plots show the semi-amortized method is significantly faster than cryoAI.

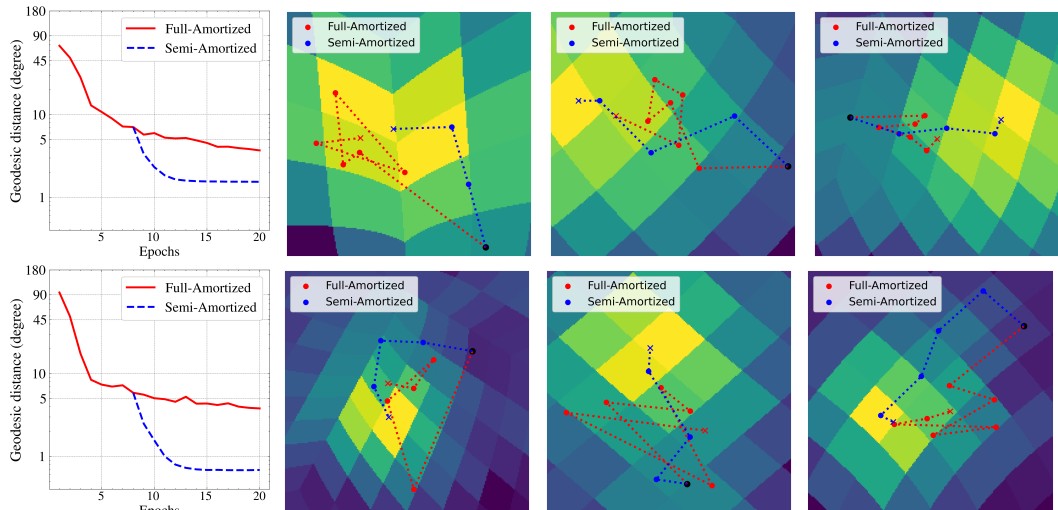

Figure 4: Quantitative and qualitative comparison of fully- vs. semi-amortized methods in pose optimization for Spike (top) and Spliceosome (bottom) datasets. **(Left)** Mean geodesic distance between the predicted pose and ground-truth is visualized at different epochs. By switching from amortized inference to direct optimization, semi-amortized method **(blue)** enjoys accelerated pose convergence compared to amortized inference **(red)**. **(Right)** To visualize pose inference, images depict the approximate log posterior for three particles (marginalized over in-plane rotations) as view-direction distribution over a HEALPix [53] uniform grid on a unit sphere $S^2$. After Gnomonic projection to 2D, we show the neighborhood of the mode of interest. **Black** dots depict starting points. **Blue** and **red** dots are pose estimates from fully- and semi-amortized methods.

In Fig. 3, we visualize resolution-time plots, showing that cryoSPIN gets to high-resolution reconstructions significantly faster than cryoAI. In fact, semi-amortization in cryoSPIN accelerates the improvement in the resolution and with our explicit structure decoder, we circumvent expensive MLP evaluations and train faster without any degradation to the final reconstruction quality as shown in Fig. 2. In Supp. B, through a detailed comparison with cryoAI, we show cryoSPIN is $\sim$ 6x faster and uses $\sim$ 5x less memory compared to cryoAI.

Finally, we report the mean and median errors in estimated poses on synthetic datasets in Table 1, showing that our method outperforms cryoSPARC and cryoAI. High errors by cryoAI on HSP dataset shows that it gets stuck in local minima and fails to accurately estimate poses.

Table 1: Mean/median errors of estimated rotations in units of degrees evaluated on synthetic datasets.

| Method | HSP | Spike | Spliceosome |
|---|---|---|---|
| CryoSPARC [8] | 6.23 / 1.05 | 1.61 / 1.51 | 1.41 / 1.36 |
| CryoAI [10] | 45.83 / 61.86 | 2.52 / 2.29 | 2.85 / 2.61 |
| CryoSPIN (Ours) | **3.27 / 0.97** | **1.54 / 0.90** | **0.68 / 0.61** |

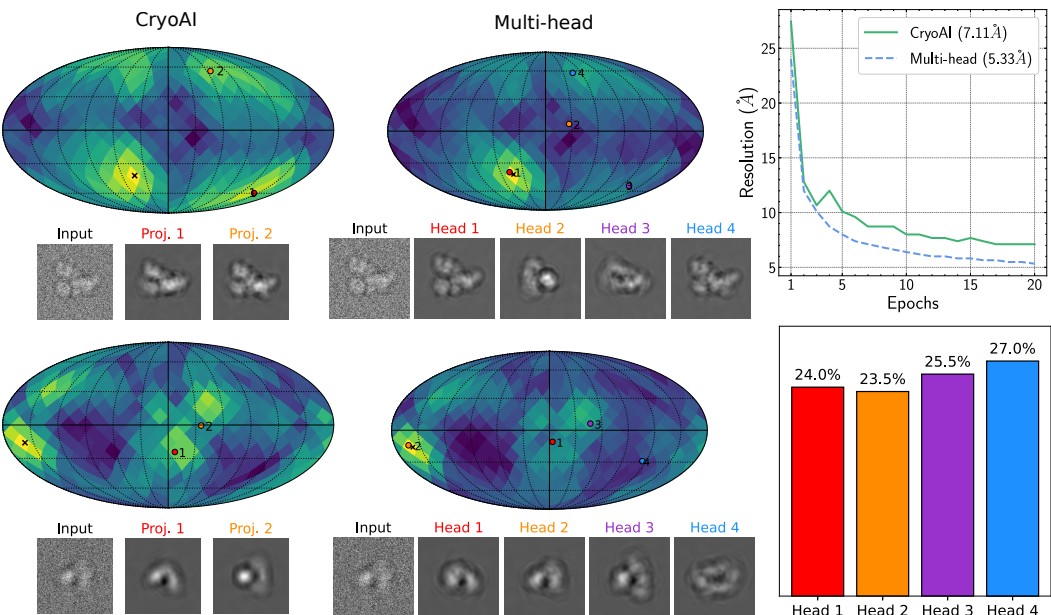

Figure 5: Comparing performance of our multi-head pose encoder ($M = 4$) with cryoAI pose encoder on the challenging HSP [45] dataset. **(Left)** The approximate log posterior of view direction is visualized on the unit sphere with highlighted areas showing modes of the distribution. CryoAI and multi-head encoders provide two and four pose estimates, respectively, which are marked with colored dots on the sphere (the order of poses is arbitrary). Below the sphere, the corresponding projections are illustrated. CryoAI fails to find the correct mode while our method is able cover multiple modes. **(Top, right)** With our multi-head encoder, the reconstruction converges to a much higher resolution compared to cryoAI. **(Bottom, right)** Percentage of images assigned to each head is visualized as a bar plot confirming that all heads participate in pose estimation.

## 5.2 Semi-Amortized vs. Fully-Amortized

To showcase the benefit of the auto-decoding stage, we compare our semi-amortized method with a fully-amortized baseline on spike and spliceosome datasets. Starting with auto-encoding, we branch the running reconstruction into two after 7 epochs: the first continues using the encoder while the second switches to direct pose optimization. We plot the average pose error with respect to the ground-truth at different epochs in Fig. 4 (left). When using the multi-head encoder, the pose with the least reconstruction error is selected among the candidates. Interestingly, as our method switches to direct pose optimization, the error in pose drops quickly, whereas the fully-amortized baseline exhibits slow convergence. This clearly shows the superiority of auto-decoding compared to auto-encoding during the later stages of optimization.

Next, we analyze the evolution of pose estimates over the optimization landscape depicted as a heat map (Fig. 4, right). Qualitatively, poses obtained by direct optimization (blue dots) show stable convergence to the optimal point (highlighted area) while those inferred by the encoder (red dots) frequently oscillate. Indeed, the encoder in amortized inference is a globally parameterized function which might be too restrictive, yielding sub-optimal pose parameters. Therefore, poses inferred in an amortized fashion might fail to consistently converge to the optimal point. On other hand, direct optimization is intuitively more flexible as it is performed separately and locally for each image, exhibiting more stable convergence. See Supp. D for more examples.

## 5.3 Multi-Modal Pose Posterior

Lastly, we examine how well cryoSPIN multi-head encoder performs vs. cryoAI encoder in terms of handling the uncertainty in the pose on the challenging dataset of HSP [45]. To simplify the visualization, we run our method with $M = 4$ heads in this experiment. We first inspect the behavior of encoders for two example cryo-EM images (Fig. 5, left, see Supp. E for more examples). In each row, the approximate posterior distribution over view direction is visualized for both cryoAI and

multi-head encoders given the input image and reconstruction. Our multi-head encoder returns a set of plausible candidates, while cryoAI obtains two pose estimates by data augmentation. In both examples, the multi-head encoder identifies the correct mode while cryoAI selects the incorrect one. Note that the pose set predicted by the multi-head encoder contains other posterior modes as well. Intuitively, cryoAI encoder with two pose predictions cannot explore the pose space as much as our multi-head encoder. Thus, the pose posterior in cryoAI remains uncertain and multi-modal during reconstruction. This also contributes to slower convergence to higher-resolution reconstructions as depicted in the top left of Fig. 5. Finally, we visualize the percentage of images assigned to each head by our "winner-takes-all" loss. The relatively even spread of assignments to different heads shows that the multi-head encoder does not suffer hypotheses collapse [30], namely all heads actively participate in the pose prediction. In Supp. C, we investigate how each head specializes in pose prediction for non-overlapping regions of $SO(3)$.

# 6 Conclusion, Limitations and Future Work

In this paper, we introduce cryoSPIN, a new semi-amortized approach to ab initio cryo-EM reconstruction. We develop a new multi-head encoder that estimates a set of plausible candidate pose to handle the uncertainty. As the uncertainty is reduced, we transition to an auto-decoding stage where poses are iteratively optimized using SGD for each image. Our results show that our multi-head encoder is able to capture multiple modes of the pose distribution, and our flexible direct optimization enables accelerated convergence of poses and reconstructions. CryoSPIN outperforms cryoAI on experimental data and achieves competitive results with cryoSPARC.

**Limitations and future work.** In this work, we assume that the 3D structure is rigid while many biomolecules are flexible in practice, exhibiting different conformations. Our results show that, even in this relatively constrained homogeneous case, there is a significant gap between amortized inference and our semi-amortized approach. We expect that the benefits of semi-amortized inference will also apply to such heterogeneous cases. In particular, our proposed multi-head encoder should be applicable to the estimation of the multimodal posterior over heterogeneity latent parameters as well as pose. Yet, there are significant challenges in heterogeneous reconstruction, such as extra uncertainty due to heterogeneity and conflation of latent spaces, making it outside the scope of our analysis on semi-amortized inference.

The development of principled criteria for switching from amortized to auto-decoding is also an interesting direction for future research. Our experiments with a range of values show that after a sufficient number of epochs, the reconstruction error becomes insensitive to the switching time. Our intuition is that when switching too early, the pose posterior may have significant uncertainty, and the pose estimates may be too far from the correct basin of attraction to afford robust convergence. Therefore, it is better to switch late than early. However, a well-defined heuristic for the optimal switch time is lacking.

**Societal impact.** Structure determination with Cryo-EM for macromolecules is one of the most exciting, fundamental areas in structural biology, and is already having with vast social impact. Cryo-EM was used to determine the structure of the SARS-COV2 spike protein, the discovery of its pre-fusion conformation, and the evaluation of medical counter-measures. It compliments advances in methods like alpha-fold for structure prediction, transforming our understanding of the process of life at the scale of the cell, and the design and development of new therapeutics.

# Acknowledgments and Disclosure of Funding

This research was supported in part by the Province of Ontario, the Government of Canada, through NSERC, CIFAR, and the Canada First Research Excellence Fund for the Vision: Science to Applications (VISTA) programme, and by companies sponsoring the Vector Institute.

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

# Supplementary Material

# CryoSPIN: Improving Ab-Initio Cryo-EM Reconstruction with Semi-Amortized Pose Inference

## A  Pose Optimization

For the auto-encoding stage, we follow cryoAI and design the convolutional backbone to output the six-dimensional representation commonly referred to as $S2S2$. To compute the rotation matrix from this representation, the 6D vector is split into two 3D vectors and normalized, denoted as $v_1, v_2 \in \mathbb{R}^3$. We then compute the cross product between them, $v_3 = v_1 \times v_2$, yielding a new unit vector. Selecting $v_1$ and $v_3$ as the first and third columns of the target rotation matrix, we compute $\tilde{v}_2 = v_3 \times v_1$, which is another unit vector orthogonal to both $v_1$ and $v_3$. Then, $R = [v_1, \tilde{v}_2, v_3]$.

Once switched to direct optimization, we change parameterization to axis-angle representation. During auto-decoding, we alternate between five iterations of pose SGD updates and one iteration of volume update. To update poses, we keep the volume fixed and optimize for the negative log-likelihood (Eq. 3) with respect to the pose parameters. We define the new pose estimate based on the current one as follows:

$$R_{t+1} = R_\delta R_t \tag{9}$$

where $R_\delta$ is an infinitesimal rotation matrix perturbing the current estimate. The perturbation matrix $R_\delta$ is parameterized by axis-angle representation. By a single vector $\boldsymbol{\omega} \in \mathbb{R}^3$, one can represent both the axis $\|\boldsymbol{\omega}\|$ and the angle $0 < \frac{\boldsymbol{\omega}}{\|\boldsymbol{\omega}\|} < \pi$ for any given rotation. Using Rodrigues formula, the perturbation matrix $R_\delta(\boldsymbol{\omega})$ can be parameterized as a function of $\boldsymbol{\omega}$. To find the optimal $\boldsymbol{\omega}$, one can initialize it with zero vector, and then use automatic differentiation [54] in pytorch to compute the gradient with respect to $\boldsymbol{\omega}$ and make updates using Adam [49]. However, a naive implementation of the function $R_\delta(\boldsymbol{\omega})$ would lead to numerically unstable calculations of the partial derivative $\frac{\partial R_\delta}{\partial \boldsymbol{\omega}}$. In fact, there is a singularity at the zero vector, and the partial derivative involves terms that are unstable around the origin. Formally, the derivative of $i$-th column of the rotation matrix $R_\delta$ with respect to the vector $\boldsymbol{\omega}$ is [55, 56],

$$\begin{aligned}
\frac{\partial R_\delta^{(i)}}{\partial \boldsymbol{\omega}} = & -\left( \mathbf{e}^{(i)} \otimes \boldsymbol{\omega} + [\mathbf{e}^{(i)}]_\times \right) \frac{\sin(\|\boldsymbol{\omega}\|)}{\|\boldsymbol{\omega}\|} + [(\boldsymbol{\omega} \cdot \mathbf{e}^{(i)})I + \boldsymbol{\omega} \otimes \mathbf{e}^{(i)}] \left( \frac{1 - \cos(\|\boldsymbol{\omega}\|)}{\|\boldsymbol{\omega}\|^2} \right) \\
& + (\boldsymbol{\omega} \otimes \boldsymbol{\omega}) \left( (\boldsymbol{\omega} \cdot \mathbf{e}^{(i)}) \frac{2\cos(\|\boldsymbol{\omega}\|) - 2 + \|\boldsymbol{\omega}\|\sin(\|\boldsymbol{\omega}\|)}{\|\boldsymbol{\omega}\|^4} \right) \\
& + [(\boldsymbol{\omega} \times \mathbf{e}^{(i)}) \otimes \boldsymbol{\omega}] \frac{\|\boldsymbol{\omega}\|\cos(\|\boldsymbol{\omega}\|) - \sin(\|\boldsymbol{\omega}\|)}{\|\boldsymbol{\omega}\|^3} .
\end{aligned}$$

where $\otimes$ and $\times$ are tensor and cross products, respectively. $\mathbf{e}^{(i)}$ is the $i$-th standard basis in 3D and $[\mathbf{v}]_\times$ denotes the cross product matrix for the vector $\mathbf{v}$. In all four terms, there are scalars such as $\frac{\sin(\|\boldsymbol{\omega}\|)}{\|\boldsymbol{\omega}\|}$ or $\frac{1 - \cos(\|\boldsymbol{\omega}\|)}{\|\boldsymbol{\omega}\|^2}$ that evaluate to $\frac{0}{0}$ at zero angle $\boldsymbol{\omega} = 0$. Similar to [55], for $\|\boldsymbol{\omega}\| \ll 1$, we substitute these terms with their numerically robust Taylor expansion, for instance,

$$\frac{\sin(\|\boldsymbol{\omega}\|)}{\|\boldsymbol{\omega}\|} = 1 - \frac{\|\boldsymbol{\omega}\|^2}{6} + O(\|\boldsymbol{\omega}\|^4) ,$$

$$\frac{1 - \cos(\|\boldsymbol{\omega}\|)}{\|\boldsymbol{\omega}\|^2} = \frac{1}{2} - \frac{\|\boldsymbol{\omega}\|^2}{24} + O(\|\boldsymbol{\omega}\|^4) .$$

We implement a differentiable and numerically stable version of the function $R_\delta(\boldsymbol{\omega})$ in pytorch and use it in our pose estimation module.

## B  Ablation Study on Decoder

We perform an ablation study on the decoder, detailed in Table 2, as well as a comparison with baselines in terms of reconstruction time per epoch, GPU memory usage, and number of parameters. As our method consists of two stages, we report numbers for each stage separately. In the first

Table 2: The reconstruction time per epoch, GPU memory, and number of parameters for different methods averaged across three runs. GPU memory is recorded separately for the encoding and decoding modules. Also, two numbers provided for semi-amortized method corresponds to auto-encoding and auto-decoding stages, respectively. In CryoAI-explicit, the implicit decoder of CryoAI is replaced with an explicit decoder. Since we store rotations in 3D representation during direct optimization, the number of parameters of pose module is $N \times 3$ with $N$ denoting number of images in millions.

| Model | Time (s) | GPU Mem. (GB) | | # Params (M) |
| | | Encoding | Decoding | |
|---|---|---|---|---|
| Semi-Amortized | 99.76, 81.61 | 1.17, - | 2.22, 0.35 | 13.01, 4.29 + $N \times 3$ |
| Fully-Amortized | 99.75 | 1.17 | 2.22 | 13.01 |
| CryoAI-explicit | 142.40 | 2.32 | 0.75 | 9.05 |
| CryoAI | 622.39 | 2.78 | 14.05 | 5.09 |

stage, the encoder has H=7 heads, while in the second stage, it is replaced with a pose module with size depending on the number of particles ($N$). To facilitate comparison, we include another baseline, cryoAI-explicit, in which the implicit decoder is replaced by an explicit one as in our method. The explicit and implicit decoders have 4.29 and 0.33 million parameters, respectively. Despite a larger decoder, our method is  6x faster and uses  5x less memory compared to cryoAI. Importantly, swapping the implicit decoder with an explicit one (cryoAI-explicit) significantly drops time and memory, indicating that the implicit decoder is a major computational and memory bottleneck. Yet, cryoAI-explicit uses  2x more GPU memory for encoding and is  1.5x slower than our method as it performs early input augmentation and runs the entire encoder twice per image. Our method, by augmenting the encoder head, saves memory and time during pose encoding. Finally, the amortized baseline, which uses an explicit decoder coupled with a multi-head encoder (H=7), uses more memory in decoding (negligible vs implicit decoder) and runs slower than the direct optimization stage of the semi-amortized method.

## C  Specialization of Encoder Heads

A natural question about the multi-head encoder is: how each head does take part in pose encoding process? To address this, using the synthetic datasets, we conduct an experiment with our multi-head architecture ($M = 4$) and visualize the performance of each head on different regions of $SO(3)$ space. In particular, as before, we define a uniform grid on the unit sphere using HEALPix [53] and assign images to their corresponding cells based on the view-direction. Now, for all images end up in the same cell, we compute the average rotation error and visualize it separately for each head. As shown in Fig. 6, all heads actively participate in pose estimation and they are able to specialize in prediction of poses for images with certain view-direction. A similar result has been provided in prior work on MCL [27, 13], to show that minimizing the error made by the best prediction ("oracle" loss) encourages diversity in deep ensembles. In our problem, by optimizing a "winner-takes-all" loss, the whole burden of pose estimation is no longer on a single network but it gets divided between multiple heads as separate predictors.

## D  More on Semi-Amortized vs. Fully-Amortized

To validate the advantages of direct pose optimization in our semi-amortized method, we further show more qualitative examples of paths taken by pose estimates over the optimization landscape during reconstruction in Fig. 7. For both methods, optimization start from the same point marked by **black** dot in the vicinity of the distribution mode. It will then continue in paths colored in **blue** and **red** for semi-amortized and full-amortized methods, respectively. We observe in all examples that iterative updates by stochastic gradient descent demonstrate a stable convergence toward the optima while poses obtained by amortized inference show unstable behavior around the mode.

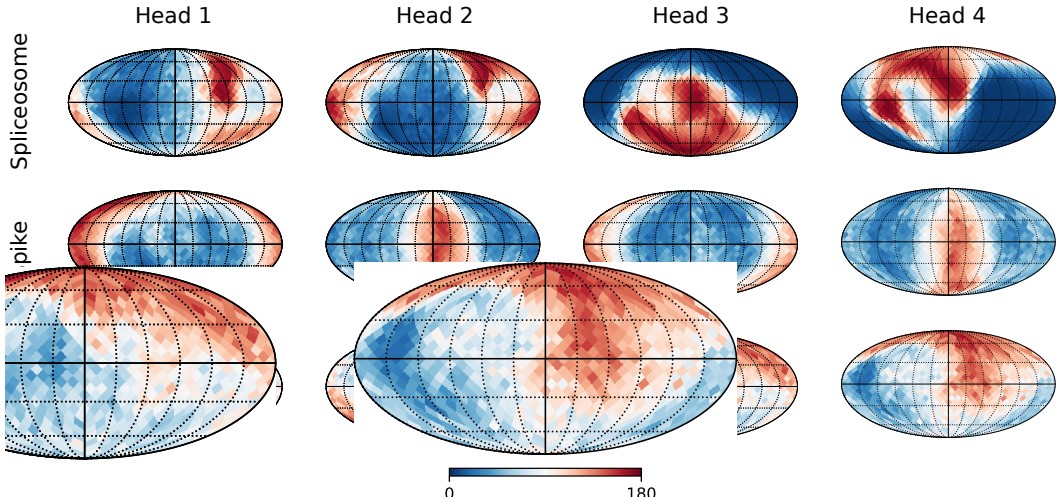

Figure 6: Average rotation error is visualized over the sphere for different heads of the multi-head pose encoder ($M = 4$). The sphere is uniformly divided into cells using HEALPix [53]; images are assigned to cells based on ground-truth view-direction. For each cell, the average rotation error is visualized, showing diverse behavior of different heads across the space. Blue and red colors show low and high error regions, respectively. Error ranges from zero to 180 degrees.

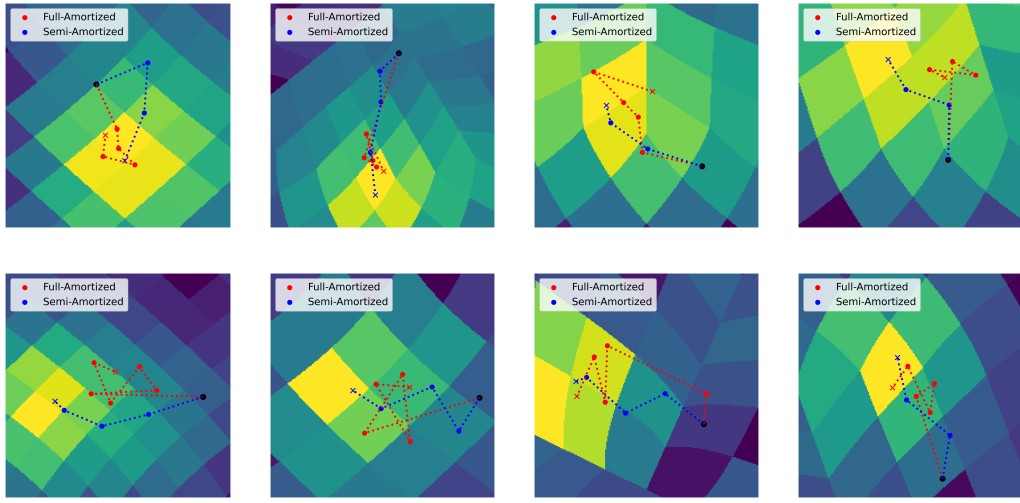

Figure 7: We compare the behavior of fully-amortized and semi-amortized pose inference on four examples per dataset. Two rows correspond to Spike and Spliceosome datasets, respectively. Each plot shows the approximate log pose posterior, marginalized over in-plane rotations represented as a heat map on a uniform grid over the unit sphere $S^2$. Gnomonic projection to 2D is also applied, followed by zooming on the proximity of the mode of interest. **Black** cross is the starting point while **blue** dots and **red** dots show poses estimated by fully- and semi-amortized methods, respectively.

# E   More on Multi-Modal Pose Posterior

Through more examples (Fig. 8), we demonstrate that cryoAI fails to handle ambiguity in pose estimation on HSP dataset. The visualization shows that pose estimates by cryoAI become stuck in incorrect modes whereas our pose encoder with multi-head architecture is able to return a pose candidate that captures the correct mode.

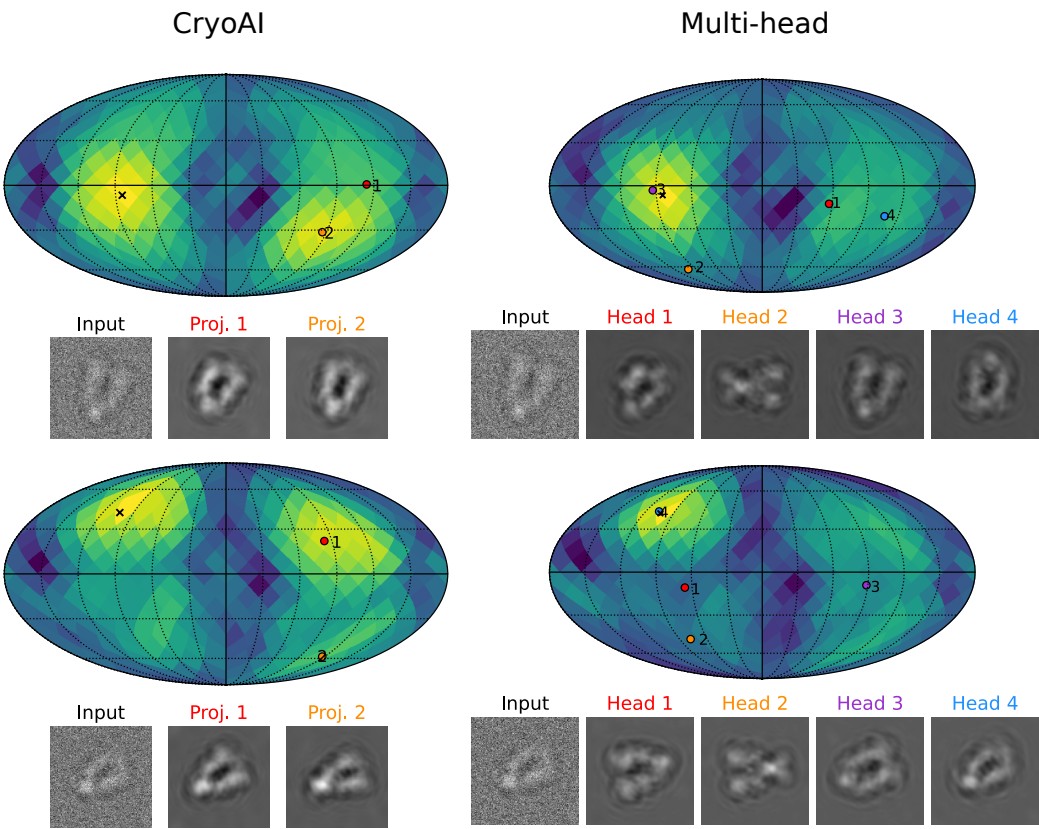

Figure 8: The approximate log posterior of view-direction visualized on the unit sphere with high-lighted areas showing modes of the distribution. CryoAI [10] and our multi-head encoders provide two and four pose estimates, respectively, which are marked with colored dots on the sphere (the order of poses is arbitrary). The corresponding projections are also illustrated. CryoAI cannot identify the correct mode of pose distribution.

