# OpenReview forum: "CryoSPIN: Improving Ab-Initio Cryo-EM Reconstruction with Semi-Amortized Pose Inference"
_NeurIPS.cc/2024/Conference — NeurIPS 2024 poster_

### Official Review · Reviewer_xz2Y · 2024-07-09

**Soundness:** 3
**Presentation:** 3
**Contribution:** 2
**Rating:** 6
**Confidence:** 4

**Summary:**

This paper introduces an approach to ab-initio homogeneous reconstruction that handles multi-modal pose distributions with a tailored encoder and accelerates pose optimization with semi-amortization. The approach uses a shared CNN feature extractor with multiple pose predictor heads, predicting several plausible poses for each image to account for pose uncertainty. Unlike the computationally expensive implicit networks used by cryoDRGN and cryoAI, this method employs an explicit 3D decoder, speeding up reconstruction. The encoder-decoder architecture is trained with a "winner-takes-all" loss, where the 3D decoder generates multiple 3D-to-2D projections, and the one with the lowest reconstruction error determines the loss. This method achieves faster convergence and more accurate poses compared to relying solely on the encoder's predictions. Evaluations show that the approach outperforms cryoAI and is competitive with cryoSPARC on both synthetic and experimental datasets.

**Strengths:**

- This paper proposes a two-stage ab-initio homogeneous reconstruction algorithm based on a novel multi-head transformer architecture to output multiple proposals of predicted poses. As a result, the performance on both synthetic and real datasets outperforms learning-based baselines.
- A novel winner-takes-all loss has been introduced to specialize multi-heads of the decoder. Each head of the decoder can account for the pose estimation in a local region, mitigating the burden on the decoder.
- This paper is well-structured and easy to follow, with appropriate references.

**Weaknesses:**

Despite its strengths, I see some major weaknesses in this paper:
1. It is unfair to claim "our semi-amortized method is faster than the amortized method of cryoAI" (Line 79) as there are some minor modifications, like changing the decoder from an implicit representation to an explicit volume to reduce the computational cost. An explicit density volume or a feature volume can easily replace all coordinate-based neural networks. In this paper, there is no ablation study about the representation of the decoder or report on how much training and inference time has been reduced by this replacement. Explicit volume with multi-head predictions leads to higher memory costs during training. Advanced representations that can achieve a better trade-off between computational and memory costs, like TensoRF, triplanes, and hash encodings, are not well discussed or explored.

2. The improvement of results is incremental. In Figure 2, the performance improvements on synthetic datasets are very subtle. On the experimental results, it seems cryoSPARC, which uses a traditional ab-initio method, achieves the best performance. Additionally, why do the resolutions of EMPIAR-10028's FSC curve all meet the resolution limit?

3. Translation estimation is important for a complete ab-initio model. Why does this method not support translation estimation while cryoAI is capable?

**Questions:**

Overall, I am not convinced that the proposed semi-amortized approach can fundamentally improve pose prediction results without any significant improvements on the image encoder side. All experiments conducted in this paper show only minor performance improvements or even no improvements compared to cryoSPARC, which has no deep learning but only traditional optimization, does it imply that the actual bottleneck is the weak image encoder? The image encoder (VGG in this paper) is not pre-trained on a diverse cryo-EM dataset but trained from scratch on a per-scene dataset in a self-supervised manner. How can it be ensured that the extracted features can be applied for accurate pose prediction? I believe the image encoder is also a performance bottleneck. A PCA visualization of the feature space of the encoder (or calculating the cosine similarity) would be very helpful to validate if images with similar poses can be clustered and images with different poses can be far from each other. It is important for this work to conduct a performance analysis for both the encoder and decoder to identify the actual problem. Unfortunately, I see no analysis in the current version of this paper. I am happy to improve my ratings if my main concerns (including weaknesses) can be addressed by the authors.

**Limitations:**

Yes, there is a discussion of the limitations in this paper, but I believe the limitation of the image encoder used in this work should be also discussed.

---

> ### Author Rebuttal · Authors · 2024-08-07
>
> Thanks for the insightful comments and questions. Below we address the concerns.
>
> **speed comparison and decoder ablation.** To clarify, we perform an additional study, detailed in the rebuttal PDF Tab.1. We report reconstruction time per epoch, GPU memory usage, and number of parameters for various methods. As our method consists of two stages, we report numbers for each stage individually: the first stage uses an image encoder with H=7 heads, while, in the second stage, the encoder is replaced with a pose module with size depending on the number of particles (N). For comparison, we include another baseline, cryoAI-explicit, in which the implicit decoder is replaced by an explicit one as in our method. The explicit and implicit decoders have 4.29 and 0.33 million parameters, respectively. Despite a larger decoder, our method is ~6x faster and uses ~5x less memory compared to cryoAI. Importantly, swapping the implicit decoder with an explicit one (cryoAI-explicit) significantly drops time and memory, indicating that the implicit decoder is a major computational and memory bottleneck. Yet, cryoAI-explicit uses ~2x more GPU for encoding and is ~1.5x slower than our method as it performs early input augmentation and runs the entire encoder twice per image. Our method, by augmenting the encoder head, saves memory and time during pose encoding. Moreover, the amortized baseline, which uses an explicit decoder with multi-head encoder (H=7), uses ~7x more memory in decoding (negligible vs implicit decoder) and runs slower than the direct optimization stage of our method.  We will include these results and clarify this comparison.
>
> **Hybrid representations.** We share our findings on more recent hybrid representations in Fig.1-B of the rebuttal PDF. In a simplified setting with known poses, we optimize the structure represented by TensoRF which yields spurious blobs of density. We conjecture this is due to the low-rank assumption made by these representations which assume a 3D signal can be factorized into 2D axis-aligned planar signals. While this assumption may be adequate for structured, real-world scenes, it tends to break down for cryo-EM structures. Thus factorization might reduce the capacity leading to artifacts of wrong density masses. We believe effective adoption of hybrid representation in cryo-EM context is non-trivial and requires further investigation. We will add these failure cases in the supplement.
>
> **Significance of improvement.** Our primary goal is to show that amortized inference methods can be enhanced by combining with more traditional optimization approaches to achieve the best of both worlds: (1) enabling pose space exploration in early stages with amortized inference empowered by deep-learning architectures, and (2) boost accuracy of DL-based models to obtain results competitive with cryoSPARC. As shown in Fig. 2, on synthetic datasets of Spliceosome and HSP, our method outperforms cryoAI by ~1A and ~2A in resolution, respectively, while achieving the same or better resolution to cryoSPARC. Similarly, for experimental data (see last row of Fig. 2), our approach provides substantial qualitative and quantitative improvements compared to cryoAI. Our reconstruction is also comparable with that obtained by cryoSPARC. Moreover, our results with shift estimation (rebuttal PDF, Fig.1A) shows that our method is able to outperform cryoSPARC on experimental data. We believe our work is a step towards developing DL-based methods for cryo-EM reconstruction and reducing their gap with traditional methods like cryoSPARC.
>
> **Resolution for EMPIAR-10028.** The deposited images in EMPIAR-10028 have a box size of D=360, with 1.34 A/pixel. In our experiments, for fair comparison to cryoAI (which learns a $128^3$ volume), the particle images are downsampled to D=128 before reconstruction. With downsampled images, the best possible resolution to obtain (at the Nyquist rate) is 7.54A. Both cryoSPARC and our method reach this limit as shown by the FSC plot in Fig. 2. We should note that this is a mid-level resolution.
>
> **Estimating in-plane shifts.**  As discussed in the ‘global response’ to all reviewers, we show in the Rebuttal PDF (Fig 1-A) that our semi-amortized approach extends in a straightforward fashion to estimation of rotation and translation while still outperforming the baselines.
>
> **Image encoder bottleneck.** There is evidence in the literature that image encoders for pose prediction are a bottleneck in amortized inference for cryoEM [1, 2]. Indeed, while improvements to image encoder may have benefits, accurate pose prediction from cryo-EM images is non-trivial because (1) input images are extremely noisy, resulting in pose ambiguities, and (2) the job of the encoder is to invert the decoder and predict pose—intuitively, this requires knowledge of the 3D structure and is difficult to accomplish from the input image alone as the 3D structure changes during optimization. Our work pursues an orthogonal direction: we take advantage of the encoder’s ability to rapidly converge to sufficiently accurate poses such that direct optimization can be performed. Our approach significantly improves pose prediction accuracy (see Table 1 and Fig. 4). For instance, on the Spliceosome our method reaches <1 degree error while cryoSPARC and cryoAI reach ~1.5 and ~2.5 degree error. Also, on the HSP dataset, cryoAI fails to handle pose uncertainty (see Fig. 5) leading to high error: ~45 degrees on average. Our method reaches ~3 degree error vs ~6 degrees for cryoSPARC.
>
> **References**\
> [1] Edelberg, et al. "Using VAEs to learn latent variables: Observations on applications in cryo-EM" (2023)\
> [2] Klindt, et al. "Towards interpretable Cryo-EM: disentangling latent spaces of molecular conformations" (2024)

---

> > ### Comment · Reviewer_xz2Y · 2024-08-09
> >
> > Thanks for authors' detailed rebuttal! The authors have thoroughly addressed my concerns based on my review comments. However, as shown in Figure 1 of the uploaded PDF, the performance of the proposed method appears to be similar to that of cryoSPARC. Could the authors please further elaborate on the advantages of the proposed method compared to cryoSPARC, such as faster speed, automation, or other benefits? If the performance is indeed comparable at this stage, what potential improvements could be made?

---

> > > ### Author Response · Authors · 2024-08-09
> > >
> > > We’re glad the rebuttal addressed your concerns, and we appreciate your quick response and thoughtful engagement.\
> > > Regarding your question, the semi-amortized method inherits some benefits from amortized methods like cryoAI in terms of runtime for large datasets (e.g. Fig. 3 (left) in the cryoAI paper shows that cryoAI requires less time than cryoSPARC to obtain a given resolution for large numbers of particles). And our inference in the amortized stage (see rebuttal PDF, Table 1) is faster than cryoAI due to encoder/decoder modifications. More specifically, based on the resolution-time plots in Fig 3 in our paper, our method reaches 10A resolution on the Spike and HSP datasets ~4x faster than cryoAI (in 11.0 and 5.3 minutes vs 41.7 and 31.4 minutes, respectively). This advantage over cryoSPARC is significant as cryo-EM datasets regularly contain more than 1M particles.\
> > > More generally, the cryo-EM community has been excited about deep learning mainly due to its potential to handle heterogeneous experimental data where inference of latents is critical along with pose. The fact that deep learning methods (like our semi-amortized inference) can now meet or exceed the performance of finely-honed methods like cryoSPARC, will help enable improved inference methods on heterogeneous data, going well beyond the current capabilities of cryoSPARC (e.g. with its linear methods for heterogeneity).

---

> ### Comment · Reviewer_xz2Y · 2024-08-10
>
> Thank you! I would like to improve my rating regarding to these discussions.

---

### Official Review · Reviewer_Kgto · 2024-07-10

**Soundness:** 3
**Presentation:** 3
**Contribution:** 3
**Rating:** 7
**Confidence:** 5

**Summary:**

This work introduces a new method for 3D ab initio estimation in cryo-EM using amortized inference. It relies on an existing technique which predicts the 3D rotation of a cryo-EM image using a convolutional neural network. That estimated rotation is then used together with a neural representation of the 3D volume to regenerate the projected image. The network weights (both of the pose estimation network and the 3D neural representation) are then optimized to reduce this auto-encoding loss.

The first important difference between the proposed framework and previous work is that this method produces several estimates using multiple heads and computes the loss with respect to the rotation estimates that yields the closest matching image. The second important difference is that after a few epochs of training, the pose estimation network is discarded, and the best pose estimates are refined using alternating minimization together with the 3D neural representation. The resulting algorithm is shown to perform well on both simulated and experimental datasets.

**Strengths:**

The writing of the manuscript is clear, in particular with respect to the introduction of the problem and its context as well as the presentation and interpretation of numerical results. The method seems to be well thought through and successfully mitigates the important pitfalls of the 3D ab initio reconstruction problem.

**Weaknesses:**

The exposition of the pose inference network and the 3D neural representation are quite superficial and deserve a more detailed explanation. For example, how are the 3D rotations obtained from the convolutional backbone output? Are they encoded in axis–angle parametrization? If so, how are the outputs properly constrained? Similarly, the 3D neural representation is described as leveraging the Hartley transform and somehow uses a decomposition of this transform into its mantissa and exponent. How is this relevant to the actual neural network architecture?

The work would also benefit from more extensive testing on experimental datasets beside EMPIAR-10028. This is a relatively high-SNR dataset depicting a very large molecule (an 80S ribosome) and is not representative of experimental datasets in general.

**Questions:**

– On like 54, what is meant by “we adopt an explicit paremeterization to further accelerate the reconstruction”? How does the parametrization affect convergence here?
– The idea of “semi-amortized inference” is never actually explicitly described. Presumably this means switching from amortized estimation to pose refinement at some point.
– How is L_{i,j} defined in eq. (6)?
– Why can the proposed method not handle shifts? Should this not be just a matter of adding another output to the pose estimation heads?
– For the synthetic and real datasets, the method switches from amortized inference to pose refinement after 7 and 15 epochs, respectively. How was this switching point determined and is there some way to automate this?
– The labels (Left) and (Right) appear to be switched in the caption for Figure 4.
– On line 227, “experiment” should be “experiments”.

**Limitations:**

The authors have largely addressed the limitations of the work in the manuscript.

---

> ### Author Rebuttal · Authors · 2024-08-07
>
> We appreciate your thoughtful feedback. In what follows, we address the main concerns and questions individually.
>
> **3D rotation parameterization.** In the supplement, Sec. A, we discuss the rotation parameterization and its optimization using PyTorch Autodiff.  Specifically, we follow cryoAI and design the convolutional backbone to output the six-dimensional representation commonly referred to as S2S2. To compute the rotation matrix from this representation, the 6D vector is split into two 3D vectors and normalized, resulting in the  unit vectors, $v_1, v_2$. We then compute the cross product ($v_1\times v_2$) to get another unit vector ($v_3$), which becomes the last column of the rotation matrix. Selecting v1 as the first column, we compute $\tilde{v_2}=v_3\times v_1$, which is a new unit vector orthogonal to $v_1$ and $v_3$. Then, $R = [v_1, \tilde{v_2}, v_3]$. During the auto-decoding stage, we switch to using a 3D axis-angle representation which requires fewer parameters.
>
> **"What is meant by “we adopt an explicit parameterization to further accelerate the reconstruction”?** We meant that the decoder uses an explicit volumetric representation of density values to model the 3D structure rather than a multi-layer perceptron (MLP). Compared to MLPs, using the explicit representation yields faster reconstruction, which is especially helpful in our case because the multi-head architecture requires querying the decoder multiple times for each input image. We will revise the text to better clarify the distinction between explicit and implicit representations, and the approach taken in our work.
>
> **On Hartley transform and decomposition.** We use the Hartley transform to save memory and reduce redundancy of the transformation of a real signal; compared to the Fourier transform, it represents frequency information using real values instead of complex values. One notable issue with the Fourier/Hartley coefficients is their high dynamic range across different frequencies. To account for this, we parameterize each coefficient using a  mantissa and exponent such that we actually store two values per voxel.
>
> **Extensive testing on experimental datasets.** To ensure fair comparison, we benchmark on EMPIAR-10028 which is the same dataset used in cryoAI. We believe this is an important experimental benchmark as it is widely adopted in methodology papers (e.g. cryoDRGN). Also, we acknowledge that further study on more recent experimental datasets with smaller particles and a range of SNRs is an important direction to explore in future work.
>
> **“The idea of semi-amortized inference is never actually explicitly described”** Thank you for raising this issue. Semi-amortized inference is described in the introduction (L58-65) and depicted in Figure 1. We provide more detail in Sec. 4.2 where “semi-amortized inference” is explicitly mentioned with references to prior work. In the revised paper we will provide a more detailed description of “semi-amortization” and justify its use in our case. For example, as higher frequency details are resolved, the variance of the pose posterior tends to decrease, becoming unimodal. At this point, the gap between the amortized and variational posterior is mainly determined by the error in the pose estimate (predicted mean). The encoder might be too restrictive as a globally parameterized function leading to limited prediction accuracy hindering further refinement of the 3D structure. This motivates us to stop amortization and switch to direction optimization, which is the main idea of semi-amortized inference. We will clarify these points in the revision.
>
> **Loss function.** We define $L_{i,j}$ as the negative log likelihood for the i-th image given its j-th predicted pose. This is formalized in Eq. 3 and is computed in the Fourier space in practice.
>
> **Estimating in-plane shifts.**  As discussed in the ‘global response’ to all reviewers, we show in the Rebuttal PDF (Fig 1-A) that our semi-amortized approach extends in a straightforward fashion to estimation of rotation and translation while still outperforming the baselines.
>
>
> **Switching point.** We considered the switching point as a hyper-parameter and experimented with a range of values. We found that after a sufficient amount of optimization (i.e., 7 epochs for synthetic data and 15 epochs for real data) the reconstruction error is insensitive to when the switch occurs. Our intuition is that when switching too early, the pose posterior may have significant uncertainty, and the pose estimates may be too far from the correct basin of attraction to afford robust convergence. Therefore, it is better to switch late than early. Exploring how to identify the optimal switching point is a promising direction that we will explore in future work.
>
> Finally, thank you for bringing the typos to our attention. We will incorporate all feedback into the revision.

---

> > ### Comment · Reviewer_Kgto · 2024-08-13
> >
> > I thank the reviewers for their rebuttal and will raise my score to 7.

---

### Official Review · Reviewer_diCb · 2024-07-13

**Soundness:** 2
**Presentation:** 2
**Contribution:** 2
**Rating:** 4
**Confidence:** 4

**Summary:**

The submission addresses ab-initio cryo-EM reconstruction, where both image poses and the 3D structure are estimated. The authors adopt a multihead architecture to estimate multiple poses for each image, encouraging the exploration of pose space early in the reconstruction process. They then refine poses in an auto-decoding fashion using SGD. Experiments on synthetic and experimental datasets demonstrate acceleration and resolution improvement over baseline approaches.

**Strengths:**

1. Compared to cryoAI, mapping the input image to multiple pose candidates in the auto-encoding stage can account for pose uncertainty and encourage exploration of the pose space during the initial stages of reconstruction.
2. The experimental results provided in the paper include both synthetic and real experimental data, as well as video results.
3. The proposed method offers a speed advantage over cryoAI.

**Weaknesses:**

1. The paper's technical contribution is primarily limited to a multi-head architecture for estimating multiple plausible poses.
2. The results presented for cryoSPARC are significantly worse than my experience suggests (i.e., on Splicesome or EMPIAR-10028). CryoSPARC's results should not be inferior to those obtained by the proposed approach.
3. The paper cannot handle heterogeneous reconstruction, which limits its general applicability in structural biology research.
4. Import baselines are missing: RELION, cryoFIRE, cryoDRGN-BNB and cryoDRGN2.

**Questions:**

1. The results presented for cryoSPARC are significantly worse than my experience. What is the setting for cryoSPARC? Are there any important assumptions?
2. Will the multi-head structure predict very similar poses, thereby undermining the assumption that this design encourages exploration of the pose space?
3. Can the authors provide a comparison of the training time and memory consumption between the proposed method and the baselines?

---

> ### Author Rebuttal · Authors · 2024-08-07
>
> Thank you for the thoughtful review and questions. Below, we address the concerns raised in the review.
>
> **"technical contribution limited to a multi-head architecture".**
> Our contributions include the design of a new encoder equipped with multiple heads to mitigate uncertainty in pose auto-encoding, and with it, a new “winner-takes-all” loss that encourages diversity in pose predictions. Beyond the architecture and objective, we propose semi-amortized pose inference, combining amortized pose inference and auto-decoding, which outperforms the predominant fully-amortized alternative. In particular, semi-amortized inference accelerates and stabilizes pose convergence (as illustrated in Fig. 4) leading to improved reconstruction.
>
> **CryoSPARC results.** In our experiments, for fair comparison to cryoAI (which learns a $128^3$ volume), the particle images are downsampled to D=128 before reconstruction. The deposited images in EMPIAR-10028 have a box size of D=360, with 1.34 A/pixel. The downsampling increases the pixel size to 3.77 A, and hence the best possible resolution (at the Nyquist rate) is 7.54 A. Both cryoSPARC and our method reach this limit as shown by the FSC plot in Fig. 2. We agree this is a mid-level resolution and hence atomic resolution details are not determined.
>
> **Heterogeneous reconstruction.** We intentionally focus on the homogeneous case instead of heterogeneous reconstruction; we show that even for the simpler homogeneous reconstruction problem there is a significant disparity between applying amortization in this setting (e.g., CryoAI) and semi-amortization (our proposed method, see Fig. 4). We expect that the benefits of semi-amortized inference will also apply to the heterogeneous reconstruction case. In particular, our proposed multi-head encoder should be applicable to the estimation of the multimodal posterior over heterogeneity latent parameters as well as pose. Yet, there are significant challenges in heterogeneous reconstruction, making it outside the scope of our analysis on semi-amortized inference—for example, (1) heterogeneity adds significant uncertainty and multimodality in the posterior over the latent variables, (2) changes in heterogeneous structure during optimization complicate pose estimation using amortized inference, and (3) existing methods for heterogeneous reconstruction conflate pose and conformational state, and disentangling these properties remains a challenge. We plan to investigate these lines in future work.
>
> **Other baselines.** We primarily compare with CryoAI and CryoSPARC as they are the state-of-the-art methods for ab-initio homogeneous reconstruction, and our focus is on investigating the tradeoff between amortized and semi-amortized inference. The deep-learning-based methods cryoFIRE and cryoDRGN (which adopt the auto-encoding approach similar to cryoAI) are specially designed for reconstruction of heterogeneous datasets. Still, we provide an additional comparison to cryoDRGN with the same setup as in the paper (see rebuttal PDF, Fig. 1-C). We run a homogenous ab initio reconstruction with cryoDRGN on synthetic and experimental datasets. The results show that our semi-amortized method outperforms cryoDRGN and reaches higher resolution reconstruction in most cases. We will add these results in the revision.
>
> **“Will the multi-head structure predict very similar poses?”.** The multi-head encoder does not predict similar poses. Figures 5 and 8, for example, show that different heads return different poses. Moreover, in the supplement, Fig. 6, we investigate the behavior of each head across all images by mapping their accuracy across different regions of pose space. As shown in the figure, the multi-head encoder adopts a divide-and-conquer approach where each head effectively specializes in pose estimation over localized regions with minimal overlap with other heads. This specialization behavior is a result of the winner-takes-all loss, which has been demonstrated to improve diversity in other prediction tasks as well [12, 13, 27]. Thus, the encoder is able to explore the pose space to help avoid local minima in the early stages of reconstruction.
>
> **Comparison on time and memory.** In the rebuttal PDF, we provide detailed comparison with baselines in terms of time, memory, and number of parameters (Table 1). As the semi-amortized method consists of two stages, we report numbers for each stage separately. In the first stage, the encoder has H=7 heads, while in the second stage, it is replaced with a pose module with size depending on the number of particles (N). For comparison, we include another baseline, cryoAI-explicit, in which the implicit decoder is replaced by an explicit one as in our method. The explicit and implicit decoders have 4.29 and 0.33 million parameters, respectively. Despite a larger decoder, our method is ~6x faster and uses ~5x less memory compared to cryoAI. Importantly, swapping the implicit decoder with an explicit one (cryoAI-explicit) significantly drops time and memory, indicating that the implicit decoder is a major computational and memory bottleneck. Yet, cryoAI-explicit uses ~2x more GPU memory for encoding and is ~1.5x slower than our method as it performs early input augmentation and runs the entire encoder twice per image. Our method, by augmenting the encoder head, saves memory and time during pose encoding. Finally, the amortized baseline, which uses an explicit decoder coupled with a multi-head encoder (H=7), uses more memory in decoding (negligible vs implicit decoder) and runs slower than the direct optimization stage of the semi-amortized method. We will add these performance comparisons in the revision.

---

> > ### Comment · Reviewer_diCb · 2024-08-12
> >
> > Thanks for the authors rebuttal! My understanding is that this work deals with the task of homogeneous ab-initio reconstruction, exactly the same with CryoAI. Although the results are improved compared to cryoAI, the method proposed in the paper does not have a clear advantage over cryoSPARC, let alone the ability of cryoSPARC to handle heterogeneous reconstruction.
> >
> > Can the authors provide some insights into the significance of their proposed method given cryoSPARC? What is the next step for the proposed method or more generally for deep learning methods? How long will it take for them to be applied in practical structural biology research?

---

> ### Author Response · Authors · 2024-08-13
>
> Thanks for the thoughtful response.
>
> There are several reasons to believe that deep learning (DL) methods have the opportunity to outperform cryoSPARC. Among them, cryoAI with amortized inference has been shown to be faster than cryoSPARC for large datasets; e.g. Fig. 3 (left) in the cryoAI paper shows that cryoAI requires less time than cryoSPARC to obtain a given resolution for large numbers of particles. In turn, our inference in the amortized stage is faster than cryoAI due to encoder/decoder modifications (see rebuttal PDF, Table 1); e.g. based on the resolution-time plots in Fig. 3 in our paper, our method reaches 10A resolution on the Spike and HSP datasets ~4x faster than cryoAI (in 11.0 and 5.3 minutes vs 41.7 and 31.4 minutes, respectively). This advantage over cryoSPARC is significant as cryo-EM datasets continue to increase in size, now regularly with more than 1M particles.
>
> A further advantage of DL is the potential for sample heterogeneity. With our semi-amortized inference, DL methods can now meet or exceed the performance of finely-honed methods like cryoSPARC for homogeneous reconstruction. This opens the door for DL and the inference of heterogeneity latents, going well beyond the capabilities of cryoSPARC. For context, cryoSPARC handles heterogeneity via (1) discrete 3D classification (to which homogeneous methods are easily extended), (2) 3D variability analysis, which is linear and lacks expressiveness, and (3) 3DFlex, which uses DL to learn non-linear variations, but is limited to conformational variability of large substructures. In contrast, our semi-amortized DL method can be extended to both conformational and compositional heterogeneity (a la cryoDRGN and cryoFIRE).
>
> Indeed, to realize the full potential of cryo-EM in practical structural biology research, it is essential to be able to model structural dynamics in heterogeneous datasets. There are open challenges in heterogeneous reconstruction, such as how to handle multi-modality in the posterior over the latent variables and the conflation of pose and conformational states. In this respect, extension of our multi-head encoder and semi-amortization to heterogeneity inference should be helpful, and we plan to investigate these lines in future work.

---

> > ### Comment · Reviewer_diCb · 2024-08-13
> >
> > Thanks for the response. I will consider it for my final decision.

---

### Author Rebuttal · Authors · 2024-08-07

We thank all reviewers for their thoughtful feedback. In the attached rebuttal PDF, we include results of additional experiments to clarify and address concerns raised by reviews, including additional baselines and datasets, and including empirical results using semi-amortized inference to estimate both particle rotation and translation, as requested by reviewers Kgto and xz2Y.

**Estimation of in-plane translation.**  Our paper originally focused mainly on the inference of 3D orientation as this is widely regarded to be more challenging than the inference of planar shifts per se. As such, we assumed that particles have been centered. Nevertheless it is straightforward to extend our method to estimate rotation and translation: We did so simply by allocating additional encoder heads to estimate translation parameters, as suggested in the reviews. We demonstrate the resulting model through application to experimental data with non-centered input particles. Fig. 1-A in the rebuttal PDF shows the reconstructed 3D density maps along with FSC curves on an experimental data (EMPIAR-10028), allowing comparison to cryoSPARC and cryoAI. A direct comparison of the FSC curves in Fig. 1-A shows that our semi-amortized approach outperforms cryoSPARC, and in Figure 4 of the original cryoAI paper clearly shows that cryoSPARC outperforms cryoAI on this same dataset. (Using the publicly available code for cryoAI by the authors, we also tried to replicate these results in order to show all FSC curves on the same plot.  However, in the presence of unknown planar translation, we found that cryoAI becomes trapped in local minima and hence we could not replicate the published FSC curves in the original paper.) We will add these results to the camera-ready version of the paper, along with the other changes or clarifications requested in the individual reviews.

---

### Decision · Program_Chairs · 2024-09-25

**Decision:**

Accept (poster)

**Comment:**

This paper introduces a method for 3D ab initio reconstruction in cryo-EM using amortized inference. The paper makes several important contributions to state-of-the-art deep learning approaches, and it is clear and well-written.